



# Characterization and comparison of PM$_{2.5}$ oxidative potential assessed by two acellular assays

Dong Gao[1], Krystal J. Godri Pollitt[2], James A. Mulholland[1], Armistead G. Russell[1], Rodney J. Weber[3]

[1]School of Civil and Environmental Engineering, Georgia Institute of Technology, Atlanta, GA 30332, USA
[2]Department of Environmental Health Sciences, School of Public Health, Yale University, New Haven, CT 06520, USA
[3]School of Earth and Atmospheric Sciences, Georgia Institute of Technology, Atlanta, GA 30332, USA

*Correspondence to*: Rodney J. Weber (rweber@eas.gatech.edu)

**Abstract.** The capability of ambient particles to generate *in vivo* reactive oxygen species (ROS), called the oxidative potential (OP), is a potential metric for relating particulate matter (PM) to health effects and is supported by several recent epidemiological investigations. However, studies using various types of OP assays differ in their sensitivities to varying PM chemical components. In this study, we systematically compared two health-relevant acellular OP

assays that track the depletion of antioxidants or reductant surrogates: the synthetic respiratory tract lining fluid (RTLF) assay that tracks the depletion of ascorbic acid (AA) and glutathione (GSH), and the dithiothreitol (DTT) assay that tracks the depletion of DTT. Year-long daily samples were collected at an urban site (Jefferson Street) in Atlanta during 2017 and both DTT and RTLF assays were applied to measure the OP of water-soluble PM$_{2.5}$ components. PM$_{2.5}$ mass and major chemical components, including metals, ions, and organic and elemental carbon

were also analyzed. Correlation analysis found that OP as measured by the DTT and AA depletion (OP$^{DTT}$ and OP$^{AA}$, respectively) were correlated with both organics and some water-soluble metal species, whereas that from the GSH depletion (OP$^{GSH}$) was exclusively sensitive to water-soluble Cu. These OP assays were moderately correlated with each other due to the common contribution from metal ions. OP$^{DTT}$ and OP$^{AA}$ were moderately correlated with PM$_{2.5}$ mass, with Pearson's r = 0.55 and 0.56, respectively, whereas OP$^{GSH}$ had a significantly lower correlation (r =

0.24). There was little seasonal variation in the OP levels for all assays due to the weak seasonality of OP-associated species. Multivariate linear regression models were developed to predict OP measures from the particle composition data. The models indicated that the variabilities in OP$^{DTT}$ and OP$^{AA}$ were attributed to not only the concentrations of metal ions (mainly Fe and Cu) and organic compounds, but also antagonistic metal–organic and metal–metal interactions. OP$^{GSH}$ was sensitive to the change in water-soluble Cu and brown carbon (BrC), a proxy for ambient

humic-like substances.

## 1 Introduction

Epidemiological studies have consistently reported associations between fine particulate matter (PM$_{2.5}$) and increased morbidity and mortality (Brunekreef and Holgate, 2002; Cohen et al., 2017; Lippmann, 2014; Norris et al., 1999; Pope et al., 2004; Samet et al., 2000; Sun et al., 2010; Thurston et al., 2017). The capacity of inhaled

particulate matter (PM) to elicit oxidative stress has emerged as a hypothesis to explain PM-induced adverse health





effects. Inhaled PM can directly introduce PM-bound reactive oxygen species (ROS) to the surface of the lung where they react with and deplete lung lining fluid antioxidants, or introduce redox-active PM species which can react with biological reductants and generate ROS *in vivo* (Lakey et al., 2016). The latter can occur in organs beyond the lungs by particles or chemical species being translocated from the lungs throughout the body. Oxidative stress

arises when the presence and production of ROS overwhelms the antioxidant defenses, and can lead to cell and tissue damage and induction of chronic and degenerative diseases (Das, 2016; Halliwell, 1994; Pizzino et al., 2017). The ability of PM to generate ROS *in vivo,* referred to as the oxidative potential (OP) of particles, has gained increasing attention as possibly a more integrative health-relevant measure of ambient PM toxicity than $PM_{2.5}$ mass which may contain a mix of highly toxic (e.g. polycyclic aromatic hydrocarbons (PAHs), quinones, and transition

metals) to relatively benign (e.g. sulfate and ammonium nitrate) PM components (Frampton et al., 1999; Lippmann, 2014).

A variety of acellular assays have been developed to assess PM OP (Ayres et al., 2008; Bates et al., 2019). In general, these assays involve the incubation of PM extracts or suspension with chemical reagents/probes, and the response is recorded over time or after incubation. The responses recorded include the depletion of reductant

surrogate, such as the dithiothreitol (DTT) assay (Cho et al., 2005), and depletion of specific antioxidants in a composite solution, such as a synthetic respiratory tract lining fluid (RTLF) model (Godri et al., 2011; Mudway et al., 2004; Zielinski et al., 1999). In contrast, other assays measure ROS generation (e.g., the dichlorofluorescein assay (Huang et al., 2016; Venkatachari et al., 2005)), or hydroxyl radical formation in the presence of $H_2O_2$ (e.g., electron paramagnetic/spin resonance (EPR/ESR) (Shi et al., 2003a; Shi et al., 2003b)). The assays based on

exposing PM species to antioxidants are currently more commonly used. The DTT assay (Cho et al., 2005) is a chemical system that mimics the *in vivo* PM-catalyzed electron transfer process. In this assay, DTT acts as a surrogate of the cellular reductant (NADH or NADPH), donating electrons to oxygen and producing ROS with the catalytic assistance of PM redox-active species. PM OP (i.e., $OP^{DTT}$ in this case) is determined by measuring the depletion of DTT over time, which is assumed to be proportional to the concentration of redox-active compounds in

PM. For the RTLF assay, RTLF is constructed to simulate the aqueous environment that particles first encounter when inhaled into the lungs and deposited. The antioxidants in RTLF, specifically the major low-molecular-weight antioxidants, ascorbic acid (AA), uric acid (UA), and reduced glutathione (GSH), provide protective defenses against PM-induced oxidative damage. The extent to which they are depleted by PM over time reflects a direct measure of PM oxidative activity by this assay, expressed as $OP^{AA}$, $OP^{UA}$, and $OP^{GSH}$ (Kelly et al., 1996; Mudway et

al., 2004; Zielinski et al., 1999). Some studies (DiStefano et al., 2009; Fang et al., 2016; Janssen et al., 2015) have used a simplified alternative approach to AA analysis in RTFL, where only the single antioxidant AA is contained in the solution. If not explicitly stated, $OP^{AA}$ in this paper represents $OP^{AA}$ obtained from the RTLF model.

Recently, a number of epidemiological studies have used some of these OP assays to examine the linkage between particle OP and adverse health outcomes. $OP^{DTT}$ of ambient fine particles has been found to be more strongly

associated with multiple cardiorespiratory outcomes, such as airway inflammation (Delfino et al., 2013; Janssen et al., 2015; Yang et al., 2016), asthma (Abrams et al., 2017; Bates et al., 2015a; Fang et al., 2016; Yang et al., 2016)





and congestive heart failure (Bates et al., 2015a; Fang et al., 2016), than PM mass. Multiple population-scale studies conducted in Canada employed the RTLF assay to assess OP of $PM_{2.5}$, and found that $OP^{GSH}$ was associated with lung cancer, cardiometabolic mortality (Weichenthal et al., 2016a), emergency room visits for respiratory illness

(Weichenthal et al., 2016c), and myocardial infarction (Weichenthal et al., 2016b). The association between airway inflammation in asthmatic children and $OP^{GSH}$ of $PM_{2.5}$ personal exposure was also reported (Maikawa et al., 2016). However, a human exposure study in the Netherlands did not find an association between $OP^{GSH}$ and acute airway inflammation after 5 h of exposure (Strak et al., 2012). A population-scale study in London found no association between $OP^{GSH}$ and mortality and hospital admission (Atkinson et al., 2016). For the AA depletion by PM assay,

either in the composite RTLF model or in a simplified AA-only model, no association with adverse health end points has been found, including cause-specific mortality and cardiorespiratory emergency department visits (Fang et al., 2016; Weichenthal et al., 2016a; Weichenthal et al., 2016b; Weichenthal et al., 2016c). Bates et al. (2019) provide a review of the relationships of various OP assays with adverse health effects.

The differences in observed health effects may be due to the different sensitivity of OP assays to various PM

species. Past studies have shown that specific assays are correlated with different PM components. $OP^{DTT}$ has been found to be sensitive to transition metals (Charrier and Anastasio, 2012; Fang et al., 2016; Verma et al., 2015a) and organic species, especially more oxygenated aromatic organics, such as quinones and hydroxyquinones (Cho et al., 2005; Kumagai et al., 2002; McWhinney et al., 2011; Verma et al., 2015b). $OP^{AA}$ obtained from the simplified AA-only model are mostly responsive to the metal content of PM (Fang et al., 2016; Yang et al., 2014). Antioxidants

(AA and GSH) within the synthetic RTLF are responsive to a slightly different group of metals. For example, $OP^{AA}$ responds to iron and $OP^{GSH}$ is related to aluminum (Godri et al., 2010). But both $OP^{AA}$ and $OP^{GSH}$ are sensitive to copper (Ayres et al., 2008). Studies performed on real PM samples or standard solutions indicate that quinones also drive the oxidative losses of both antioxidants (Ayres et al., 2008; Calas et al., 2018; Kelly et al., 2011; Pietrogrande et al., 2019).

Since different assays capture different chemical fractions of the oxidative activity of PM, it is challenging to synthesize the findings from OP-health studies. There remains a need to compare different assays on identical particle samples to advance our understanding of the effects of PM species on OP measures, and in turn assess the results of the health studies that use OP. In this study, we used two acellular OP assays, DTT and RTLF, to measure the water-soluble OP of ambient $PM_{2.5}$ collected from urban Atlanta over one-year period. These two assays were

chosen since they are currently most commonly used and have shown significant associations with adverse health outcomes in some studies. A suite of chemical components was also measured on these samples and univariate and multivariate linear regression analyses were performed to identify and evaluate the contribution of major chemical components to each of these OP metrics.

## 2 Methods

### 2.1 Sampling

Year-long sampling was conducted in 2017 from 1 January to 30 December at the Jefferson Street SEARCH (Southeastern Aerosol Research and Characterization) site (Edgerton et al., 2006, 2005). Jefferson Street is situated



roughly 4.2 km northwest of downtown Atlanta and 2.3 km from a major interstate highway and is representative of urban Atlanta region. The site has been extensively used in past studies characterizing urban-Atlanta air quality

(Hansen et al., 2006) and the data used in OP and epidemiological studies (Abrams et al., 2017; Bates et al., 2015b; Fang et al., 2016; Sarnat, 2008; Verma et al., 2014).

Ambient fine particles were collected daily (from midnight to midnight, 24 h integrated samples) onto pre-baked 8×10 in. quartz filters (Pallflex Tissuquartz, Pall Life Sciences) using high-volume samplers (HiVol, Thermo Anderson, nominal flow rate 1.13 $m^3$ $min^{-1}$, $PM_{2.5}$ impactor). A total of 349 filter samples were collected for

analysis; missing days were due to instrumentation issues. The HiVol quartz filters were wrapped in pre-baked aluminum foil after collection and stored at -18 °C until analyses. $PM_{2.5}$ mass concentration was monitored continuously by a tapered element oscillating microbalance (TEOM, Thermo Scientific TEOM 1400a), the sample stream dried at 30 °C using a Sample Equilibration System (Meyer et al., 2000). A Sunset semi-continuous OCEC analyzer (Sunset Laboratory) was used to provide in situ measurements of organic and elemental carbon (OC/EC)

content of fine PM. The data were obtained hourly by using 1 h cycles in which the instrument sampled ambient air through an activated carbon denuder for 45 min and analyzed the particles collected on the quartz filter for 15 min using NIOSH 5040 analysis protocol (Birch and Cary, 1996).

### 2.2 Oxidative potential measurements

Two acellular assays, DTT and RTLF assays, were performed to measure the oxidative potential of water-soluble

$PM_{2.5}$. The DTT analysis was completed at Georgia Institute of Technology, and all filters were analyzed within one month after collection. The RTLF assay was conducted at Yale University during Oct. 2018. Prior to the OP analyses, a fraction of each HiVol filter (5.07 $cm^2$ for DTT and 4.5 $cm^2$ for RTLF) was punched out, placed in a sterile polypropylene centrifuge vial (VWR International LLC, Suwanee, GA, USA) and then extracted in 5 mL of deionized water (DI, >18 MΩ $cm^{-1}$) via 30 min sonication. The water extract was filtered through a 0.45 µm PTFE

syringe filter (Fisherbrand, Fisher Scientific) and then used for OP analysis.

### 2.2.1 DTT assay

The DTT assay was performed with a semi-automated system developed by Fang et al. (2015b), following the protocol described by Cho et al. (2005). In brief, the PM extract (3.5 mL) was incubated with DTT solution (0.5 mL; 1 mM) and potassium phosphate buffer (1 mL; pH~7.4, Chelex-resin treated) at 37 °C. A small aliquot (100 µL) of

the mixture was withdrawn at designated times (0, 4, 13, 23, 31 and 41 min) and mixed with trichloroacetic acid (TCA, 1 % w/v) to quench the DTT reactions. After addition of Tris buffer (pH ~8.9), the remaining DTT was reacted with 5,5'-dithiobis-(2-nitrobenzoic acid) (DTNB) to form a colored product which absorbs light at 412 nm. The final mixture was pushed through a 10 cm path length liquid waveguide capillary cell (LWCC; World Precision Instruments, Inc., FL, USA), and the light absorption was recorded by an online spectrometer, which included a UV-

Vis light source (DT-mini-2, Ocean Optics, Inc., Dunedin, FL, USA) and a multi-wavelength light detector (USB4000 Miniature Fiber Optic Spectrometer). The DTT consumption rate, used as a measure of OP, was determined from the slope of the linear regression of DTT residual vs. time. Good linearity was found for all samples with correlation coefficients ($R^2$) larger than 0.98. In parallel with all sample batches, at least one field



blank and one positive control (9,10-phenanthraquinone) was analyzed, and their OP values remained constant

throughout the analysis. The PM OP measured by this assay (i.e., $OP^{DTT}$) was blank-corrected and normalized by the air volume that passed through the extracted filter fraction, expressed as nmol DTT $min^{-1}$ per $m^3$. This approach did not involve the use of PM samples with constant mass, which is sometimes employed to limit nonlinear DTT response to certain metal ions (Charrier et al., 2016).

**2.2.2 RTLF assay**

The RTLF assay is based on the protocol adopted by Maikawa et al. (2016). PM water extracts were transferred into a 96-well microplate with 180 µL of sample liquid in each well. 20 µL of synthetic RTLF (pH ~7.0) containing equimolar concentrations (2 mM) of AA, UA and GSH was added into each well, resulting in a final starting concentration of 200 µM of antioxidants. The PM-RTLF mixture was incubated in a plate reader (SpectraMax190, Molecular Devices, LLC, San Jose, CA, USA) for 4 hours at 37 °C with gentle mixing. Following incubation, the

concentrations of AA and GSH were analyzed immediately. UA concentration was not measured since studies have consistently suggested that no depletion of UA was observed in the presence of PM (Kunzli et al., 2006; Mudway et al., 2004; Zielinski et al., 1999).

AA concentration was determined with the plate reader by measuring the light absorbance at 260 nm. The GSH concentration was indirectly quantified by measuring total glutathione (GSx) and oxidized glutathione (GSSG)

concentrations, both compounds determined using the enzymatic recycling method (Baker et al., 1990). The incubated PM-RTLF mixture was diluted 49-fold with 100 mM sodium phosphate buffer (pH ~7.5) containing ethylenediaminetetraacetic acid (EDTA). To measure the GSx concentration, 50 µL of each diluted sample was dispensed onto a microplate. 100 µL of reaction mixture (0.15 mM DTNB, 0.2 mM NADPH and 1 U glutathione reductase (GR)) was added to each well. In the mixture, GSH reacted with DTNB, forming a yellow colored product

5-thio-2-nitrobenzoic acid (TNB) and the mixed disulfide GS-TNB. In the presence of NAPDH and GR, GSSG and GS-TNB were reduced back to GSH, leading to more TNB production. The plate was analyzed on the plate reader for two minutes under constant mixing to continuously monitor the formation of TNB. The TNB formation rate, which is proportional to the GSH concentration, was measured at an absorbance of 405 nm. For GSSG measurement, 5 µL of 2-vinly pyridine (2-VP) was added to 130 µL of the diluted sample to conjugate GSH. The

solution was incubated at room temperature for 1 hour, followed by similar procedures performed for the GSx measurement. The GSH concentration was calculated by subtracting two times the GSSG concentration from the measured GSx concentration.

Field blanks and known controls (e.g. positive controls: $H_2O_2$ and Cu; negative control: Zn) were run in parallel with all sample batches. All samples and controls were measured in triplicate. The percentage of AA and GSH depletion

after 4 h incubation for each PM sample was calculated relative to the field blank. PM OP obtained from this assay, i.e., $OP^{AA}$ and $OP^{GSH}$, was determined by normalizing the percentage loss with the sampled air volume, in unit of % depletion per $m^3$.





### 2.3 Chemical analysis on PM filters

#### 2.3.1 Elemental analysis

Both total and water-soluble trace metals were determined by inductively coupled plasma-mass spectrometry (ICP-MS) (Agilent 7500a series, Agilent Technologies, Inc., CA, USA), including magnesium (Mg), aluminum (Al), potassium (K), calcium (Ca), chromium (Cr), manganese (Mn), iron (Fe), copper (Cu) and zinc (Zn). For the determination of concentrations of total metals, a 1.5 cm$^2$ filter punch from the HiVol quartz filter was acid-digested for 20 min using aqua regia (HNO$_3$+3HCl). The acid-digested sample was then diluted in DI water to 10 mL, filtered

with a 0.45 µm PTFE syringe filter. For the analysis of water-soluble metals, no digestion was required. In this case, one circular punch (1 in. diameter) was extracted in 5 mL of DI via 30 min sonication. The extract was filtered using a 0.45 µm PTFE syringe filter, and then acid-preserved by adding concentrated nitric acid (70 %) to a final concentration of 2 % (v/v).

#### 2.3.2 Water-soluble organic carbon (WSOC) and brown carbon (BrC)

Two 1.5 cm$^2$ filter punches from the HiVol filter were extracted in 15 mL DI in a pre-baked glass centrifuge vial (DWK Life Sciences, Rockwood, TN, USA) by 30 min sonication. The extracts filtered with 0.45 µm PTFE syringe filters were used to measure water-soluble organic carbon (WSOC) and its light absorption properties (BrC, used as a source tracer). A fraction (~6 mL) of filter extract was injected by a syringe pump (Kloehn, Inc., NV, USA) into a 2.5 m path length LWCC (World Precision Instruments, Inc., FL, USA), with an internal volume of 500 µL. The

absorbance at 365 nm wavelength (BrC) was measured by an online spectrophotometer. The remaining liquid extract was drawn into Sievers total organic carbon (TOC) analyzer (Model 900, GE Analytical Instruments, Boulder, CO, USA) for determination of WSOC concentration. The TOC was calibrated with a series of prepared sucrose standards.

#### 2.3.3 Water-soluble ionic species

One 1.5 cm$^2$ filter punch from the HiVol filter was extracted in 10 mL DI via sonication. The inorganic ions (SO$_4^{2-}$, NO$_3^-$, and NH$_4^+$) in the filtered water extracts were measured by ion-exchange chromatography (IC) with conductivity detection (Anion: Metrosep A Supp 5–150/4.0 anion separation column; eluent: 3.2 mM Na$_2$CO$_3$, 1.0 mM NaHCO$_3$, eluent flow rate 0.78 mL/min. Cation: Metrosep C 4–150/4.0 cation separation column; eluent: 1.7 mM HNO$_3$, 0.7 mM dipicolinic acid, eluent flow rate 0.9 mL/min) with an automated sampler (Dionex AS40,

Thermo Fisher Scientific, Waltham, MA).

### 2.4 Multivariate regression models

Multivariate linear regression models were developed to predict OPs based on PM speciation data and investigate the relative importance of species on different OPs. Prior to the regression analysis, the boxplots were used to identify the outliers and test the normality of data. The extreme values (a total of around 3 % of the OP

measurements) were removed from the data set. Linear regression was performed between PM components and the various OPs. To simplify the analysis, PM components correlated with OP (r>0.4, p<0.05) were selected as the independent variables of the models. A stepwise regression was applied to the data set using Matlab R2016a to form the multivariate regression models. To evaluate the performance of the final models, 5-fold cross validation was





employed and repeated 50 times. For each OP measure, the average mean-squared error over 50 iterations was within 25 % of the mean OP value (23.4 %, 17.9 % and 12.2 % for $OP^{DTT}$, $OP^{AA}$ and $OP^{GSH}$, respectively).

### 3 Results and discussion

#### 3.1 Ambient PM composition

Figure 1 shows the time series of $PM_{2.5}$ mass concentration and the averaged chemical composition of ambient particles collected at the site. A factor of 1.6 was applied to convert organic carbon to organic matter (Turpin and

Lim, 2001; Weber, 2003). Reconstituted mass from measured chemical species agreed well with $PM_{2.5}$ mass measured by the TEOM with Pearson's r=0.84, and accounted for more than 80 % of the $PM_{2.5}$ mass. The missing mass may result from other species not measured, semi-volatile material lost from the filter, and the uncertainty in converting measured carbon mass to organic matter (factor of 1.6 used).

Fractions of various chemical components in $PM_{2.5}$ are consistent with previous observations (Edgerton et al., 2005;

Verma et al., 2014). In general, PM mass was dominated by organic compounds (WSOM+WIOM~50 %), followed by inorganic ions (10 % $SO_4^{2-}$, 4–7 % $NH_4^+$, and 1–8 % $NO_3^-$). Metals constituted 6–13 % of the PM mass, among which water-soluble metals were at trace amounts (1–2 %). EC accounted for a small fraction of the PM mass (5–6 %). $NH_4^+$, and $NO_3^-$, which are semi-volatile, accounted for a larger fraction of fine particle mass during the cold season. The metal fraction also increased in winter, whereas the fractions of other PM components did not vary

significantly during the sampling period.

Although insoluble PM components also play an important role in OP (Gao et al., 2017; Verma et al., 2012), this study solely focus on water-soluble OP measurements, and thus the water-soluble PM components are the primary focus in this study.

#### 3.2 Association of OP with PM components

Pearson's correlation coefficients for the linear regression between OP and select chemical components are shown in Table 1. The detailed correlation matrices for different seasonal periods are given in Table S1–S3. We defined the strength of the absolute correlation coefficient value as strong for values ≥ 0.65, moderate from 0.40 to 0.65, and weak for values < 0.4. The OP assays were moderately inter-correlated over the sampling year. In all cases (whole year and summer or winter), $OP^{AA}$ and $OP^{GSH}$ had the highest correlations, which may in part be due to these

measurements being conducted on the same sample extracts. As for $OP^{DTT}$, the correlations with $OP^{AA}$ and $OP^{GSH}$ varied, but were largely similar. The correlations between the OP measures and various PM components varied, highlighting the different sensitivities of OP assays to various PM components. Note these correlations do not imply that the compounds are responsible for the OP as some of them are not redox active compounds. However, they could be considered as indicators of other compounds simultaneously produced by the same source.

As shown in Table 1, $OP^{DTT}$ was correlated with OC and WSOC, indicating a contribution from PM organic compounds. The correlations between $OP^{DTT}$ and certain water-soluble metals, such as Fe, Cu and Mn, were also observed. A moderate correlation of EC with $OP^{DTT}$ (r=0.51) and somewhat with metals and OC (r=0.55, 0.43 and 0.83 for Fe, Mn and OC, respectively; Table S1) suggests that incomplete combustion could be one of their common



sources. The associations of $OP^{DTT}$ with PM species are consistent with a number of previous studies (Fang et al.,
2016; Fang et al., 2015b; Verma et al., 2014; Yang et al., 2014), though the correlations in our work were weaker
(r > 0.5 compared with r>0.65 in other studies).

Similar to $OP^{DTT}$, $OP^{AA}$ was moderately correlated with OC, WSOC and water-soluble metals, mainly Fe and Cu
(r=0.47–0.55). The results are compared with a previous study conducted in the same Atlanta region by Fang et al.
(2016) wherein a simplified AA assay was applied to assess water-soluble OP of $PM_{2.5}$. The AA depletion in the
AA-only model was found to be strongly correlated with water-soluble Cu with Pearson's r >0.65, and associations
with WSOC (or BrC) and metals were also observed. The weaker sensitivity of AA to water-soluble Cu observed in
our study is possibly due to the reactivity hierarchy existing within the antioxidant model with GSH>AA>> UA
(Zielinski et al., 1999). This is further supported by other studies. In the study of Charrier et al. (2011), ligand
speciation modeling indicated that GSH was a stronger ligand compared to AA and caused a dramatic shift in Cu
speciation by forming Cu–GSH complexes. The experimental results in the study of Pietrogrande et al. (2019)
showed that the response of the acellular AA assay was strongly dependent on the composition of synthetic RTLF
used and the presence of GSH and UA would lower the sensitivity of AA response to Cu. The correlations of
organics and EC which comprised a large fraction of PM mass (Fig. 1), with $OP^{DTT}$ and $OP^{AA}$, likely account for the
$OP^{DTT}$ and $OP^{AA}$ correlations with PM mass.

In contrast to $OP^{DTT}$ and $OP^{AA}$, $OP^{GSH}$ was found to be exclusively correlated with water-soluble Cu with Pearson's
r >0.7. The consistent lower correlation of $OP^{AA}$ with water-soluble Cu than $OP^{GSH}$ with Cu, is consistent with GSH
outcompeting AA in the RTLF in forming Cu complexes. The results are also consistent with other studies (Aliaga
et al., 2010; Ayres et al., 2008; Godri et al., 2011).

The correlations differed by seasons. In winter, $OP^{AA}$ and $OP^{DTT}$ were more correlated with organic species, with
stronger associations with WSOC, BrC and K, indicating biomass burning as a common source of $OP^{AA}$ and $OP^{DTT}$.
In summer, all OP assays tended to be metal-driven. $OP^{AA}$ and $OP^{DTT}$ were more correlated with Cu, along with
$SO_4^{2-}$, suggesting possible influence of secondary processing on metal mobilization (Fang et al., 2017; Ghio et al.,
1999) and resulting in a strong inter-correlation between different OP metrics.

### 3.3 Temporal variation

The time series of the monthly averages of different OP measures are shown in Fig. 2. Significant seasonal
variability in these OP measures was not evident; only a subtle seasonal variation observed for $OP^{DTT}$ and no
variations for $OP^{AA}$ and $OP^{GSH}$. $OP^{DTT}$ was slightly higher during the cold period (Jan–Feb and Nov–Dec) with an
average level of 0.24±0.08 nmol $min^{-1}$ $m^{-3}$ compared to 0.20±0.04 nmol $min^{-1}$ $m^{-3}$ in the warm period (May–Aug)
and a median $OP^{DTT}$ ratio between two periods of 1.20. However, $OP^{AA}$ and $OP^{GSH}$ had more similar levels across
seasons, with median ratios between cold and warm periods of 1.10 and 0.97, respectively.

The seasonality in OP measures should result from the temporal variations in PM species driving the various OP.
From the temporal variation of the OP-associated species shown in Fig. 3, BrC had an obvious seasonality, higher in
winter and lower in summer, which is due to the stronger influence of biomass burning in winter. The variation in




BrC may lead to the small variation in OP$^{DTT}$, considering the good correlation between OP$^{DTT}$ and BrC in winter.

Water-soluble Cu is slightly higher in mid-summer and water-soluble Fe is slightly higher in fall, but these trends are not seen in the various measures of OP.

### 3.4 Multivariate model

Given that one or more PM components contributed to these measures of OP, multivariate linear regression analysis was conducted to identify the main water-soluble PM components that drive the variability in OP and provide a

contrast between the assays. Water-soluble organic species (WSOC or BrC) and metals, mainly Fe, Cu, Mn, were selected as the independent variables to form multivariate linear regression models for OP$^{DTT}$ and OP$^{AA}$, based on their high correlations, as noted above. For OP$^{GSH}$, WSOC and BrC were used as input in addition to water-soluble Cu to include the possible influence of organic species on OP$^{GSH}$. The resulting linear relationships between different OP measures and PM components are shown in Table 2. The time series of measured and predicted OPs

and the contributions of model variables to each OP measure are given in Fig. S1. Overall, the multivariate models explained variability in OP measures reasonably well with the coefficients of determination between modeled and measured OPs (R$^2$) greater than 0.4, with the models better capturing the OP$^{AA}$ and OP$^{GSH}$ variability. In the regression results for OP$^{DTT}$ and OP$^{AA}$, components including water-soluble Fe, Cu and BrC (or WSOC) and interaction terms between metal–organic and metal–metal were included, suggesting that the variability of OP$^{DTT}$ or

OP$^{AA}$ is dependent upon not only bulk concentrations of PM components but also interactions between species. The regression model for OP$^{GSH}$ captured the contributions from Cu and BrC but had no interaction terms. The intercept in each regression model, though large and accounting for over 50 % of the mean of OP measures (Fig. S1), is practically meaningless, because the regression models are applicable only when the PM components are at ambient concentrations.

Note that BrC and WSOC are present in different models. Although they were correlated with each other (Table S1–S3), and both represent the contribution from organic species, there is a difference between these two parameters. It has been found that BrC predominantly represents the hydrophobic organic fraction (i.e., the humic-like substances (HULIS) fraction) in PM (Verma et al., 2012) and is largely from incomplete combustion (mainly biomass burning) (Hecobian et al., 2010). For example, quinones, as a subset of the HULIS fraction (Verma et al., 2015b), can be

estimated better with BrC than with WSOC. WSOC also includes organic compounds present in the hydrophilic fraction, e.g., levoglucosan, (Lin and Yu, 2011) and low molecular weight organic acids (Sullivan and Weber, 2006) and thus is a more integrative measure of organic compounds compared to BrC. The difference between BrC and WSOC is also supported by the different seasonal variation observed in BrC and WSOC (Fig. 3).

For OP$^{DTT}$ results, the presence of Cu and BrC in the equation is as expected, since Cu and organic species have

been found active in DTT oxidation (Charrier and Anastasio, 2012; Cho et al., 2005). However, Fe, which has a low intrinsic DTT activity (Charrier and Anastasio, 2012), was found to be predictive of OP$^{DTT}$, likely suggesting that Fe represents surrogate measures of constituents with intrinsic redox active properties which were not quantified. This is also supported by the evidence that Fe had correlations with other PM constituents such as OC and EC (Table S1–S3) which may suggest that Fe in the PM water extracts is solubilized by forming complexes with combustion-





derived organic species. The interaction terms, along with their negative coefficients, suggest antagonistic
interactions between Cu and organic compounds and between Cu and Fe. The interaction between metal and
organics, though not taken into account when applying multivariate regression analysis in previous studies (Calas et
al., 2018; Verma et al., 2015a), is consistent with experimental results (Yu et al., 2018) wherein antagonistic
interactions between Cu and ambient HULIS were observed in the DTT consumption. However, the interaction

between metals contrasts with experiments which showed additive effects for metal mixtures (Yu et al., 2018). But it
should be noted that the interactions among metals were usually tested with mixtures of individual species, which
can poorly represent the complex chemistry of ambient PM.

For RTLF assay, the variability of $OP^{AA}$ was attributed to the concentrations of Fe, Cu and WSOC, antagonistic
metal–organic interaction between Fe and organic compounds and metal–metal competition between Fe and Cu.

Even though the RTLF assay in previous studies was generally used to measure the OP of methanol-extracted PM
suspension, the contributions from metals and organics observed in our water-soluble OP are in agreement with
previous results (Ayres et al., 2008; Kelly et al., 2011; Pietrogrande et al., 2019). The presence of interaction terms
is novel and is supported by little empirical evidence. The antagonistic interaction between Fe and WSOC is
reasonable. Fe in the water extracts has been found largely complexed with organic compounds (Wei et al., 2019).

Since AA is not a strong ligand for Fe (Charrier and Anastasio, 2011), the complexation between Fe and organic
compounds can prevent Fe from reacting with AA. For $OP^{GSH}$, despite weak correlation of $OP^{GSH}$ with BrC, BrC
still accounted for the variability in $OP^{GSH}$, consistent with previous findings that $OP^{GSH}$ is responsive to quinones
(Ayres et al., 2008; Calas et al., 2018).

It is noteworthy that the multivariate regression models do not account for the possible nonlinear behavior of species

with OP responses. For example, nonlinear concentration–response curves have been found for DTT oxidation by
dissolved Cu and Mn (Charrier and Anastasio, 2012), which may not be characterized in the multivariate regression
model, potentially affecting the accuracy of the $OP^{DTT}$ model. We note that the variables in the models not only
represent the contribution from individual species, but also show a possible influence from co-emitted unquantified
components. There are also interactions existing among PM species, affecting the relationships between PM

compounds and OP metrics.

To further investigate the sensitivity of different OP assays to PM species, standardized regression was applied to
rescale the variables measured in different units and make the coefficients in the regression equations comparable. It
is also an effective way to reduce collinearity induced by the inter-correlated nature of PM species and the existence
of interaction terms. The standardized coefficient of a specific component indicates the estimated change in an OP

measure for every one-unit increase in component. The higher the absolute value of the standardized coefficient, the
stronger the effect of the PM species on the OP measure.

Figure 4 shows the relative importance of each PM component to OP metrics based on the calculation of
standardized coefficients. As shown in Fig. 4, water soluble Fe was the most important variable in the model of
$OP^{DTT}$, followed by BrC, Cu and antagonistic interactions. Even though $OP^{DTT}$ is not responsive to water-soluble Fe,

Fe may be a surrogate measure of compounds co-emitted with Fe from brake/tire wear and secondary formation



which have been identified as two major sources of Fe in the southeastern US (Fang et al., 2015a). For $OP^{AA}$, the strength of the effects of Fe, Cu and WSOC on $OP^{AA}$ was similar. $OP^{GSH}$ was approximately four times more sensitive to Cu than to BrC, with standardized coefficients of 0.70 and 0.18 for Cu and BrC, respectively. These results show clear contrasts between the assays, where $OP^{DTT}$ and $OP^{AA}$ are more similar and both have significant contrasts to $OP^{GSH}$.


In all these cases, it must also be kept in mind that the measurements were performed on the PM water extracts, which are not the conditions that are found in the ambient aerosol. Thus, inferring associations between species in the extracts and applying to ambient conditions is not straightforward, however, this analysis is useful for interpreting and contrasting the possible causes for associations between these types of assays and any health effects.


Water-soluble Fe, as the most important determinant of $OP^{DTT}$, has shown the strongest estimated effect on cardiovascular outcomes in the Atlanta metropolitan region (Ye et al., 2018), which may account for associations between $OP^{DTT}$ and health outcomes observed in this region. $OP^{GSH}$ is strongly dependent on a limited number of PM components, and thus associations between $OP^{GSH}$ and health outcomes may vary more significantly by regions than other assays do and associations could be expected if the PM toxicity in a region is mainly driven by specific


species, such as water-soluble Cu. $OP^{AA}$ is affected by the composition of synthetic lung fluid, and thus the AA responses obtained from RTLF and AA-only model are not comparable and should be distinguished from each other. In the RTLF model, the response of $OP^{AA}$ metric to PM samples is diminished due to the presence of GSH, possibly leading to weaker associations between $OP^{AA}$ and health endpoints.

**4 Conclusions**


In this study, a comparison was made between two of the most common techniques used for the assessment of PM oxidative potential based on antioxidant depletion from a complex synthetic RTLF ($OP^{AA}$ and $OP^{GSH}$) and DTT oxidation ($OP^{DTT}$). These two assays were used to quantify the water-soluble OP of ambient $PM_{2.5}$ collected in urban Atlanta over a one-year period based on daily filter samples. We observed moderate correlations among the OP assays, suggesting different sensitivities of OP measures to PM species. Univariate and multivariate regression


analyses indicated that $OP^{DTT}$ and $OP^{AA}$ were correlated to organic species and water-soluble metals (Fe and Cu) and were negatively affected by the interactions among species. At a more detailed level, for organic components, $OP^{DTT}$ was associated specifically with HULIS and incomplete combustion products identified by BrC, whereas $OP^{AA}$ was associated to a more general measure of organic components, WSOC. $OP^{GSH}$, though also affected by organic species, was predominantly sensitive to water-soluble Cu. Subtle temporal variation in $OP^{DTT}$ and no


seasonal variations in $OP^{AA}$ and $OP^{GSH}$ were observed, which appears to be due to little seasonality in the combined PM constituents affecting each assay. A small $OP^{DTT}$ variation was associated with variation in BrC that was higher in the cold seasons.

This study suggests that all three OP metrics are associated with transition metal ions. However, $OP^{DTT}$ and $OP^{AA}$ are more chemically integrative OP measures compared to $OP^{GSH}$, and thus may be more informative when try to



find linkage between OP and health end points. The multivariate regression models for different OP measures
      indicate the degree to which OP variability in the PM water extracts is predicted by PM constituents.

*Data availability.* Data can be assessed by request (rweber@eas.gatech.edu).

*Author contributions.* DG collected and analyzed the data and drafted the manuscript. KJGP assisted with RTLF
      measurements. JAM and AGR helped with model development. The data were interpreted by DG and RJW. RJW
      conceived, designed and oversaw the study. All authors discussed the results and contributed to the final manuscript.

*Competing interests.* The authors declare that they have no conflict of interest.


*Acknowledgements.* This publication was made possible by Georgia Tech EAS Jefferson St Funds which was a
generous gift from Georgia Power/Southern Company. We would like to thank Linghan Zeng, Gigi Pavur, Joseph
Caggiano and Allison Weber for assisting with the sampling campaigns and the lab work.

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




**Table 1. Pearson's r between OP and PM composition**

| | Overall | | | Summer | | | Winter | | |
|---|---|---|---|---|---|---|---|---|---|
| | $OP^{AA}$ | $OP^{GSH}$ | $OP^{DTT}$ | $OP^{AA}$ | $OP^{GSH}$ | $OP^{DTT}$ | $OP^{AA}$ | $OP^{GSH}$ | $OP^{DTT}$ |
| $OP^{GSH}$ | **0.67\*\*** | | | **0.78\*\*** | | | 0.62\*\* | | |
| $OP^{DTT}$ | 0.61\*\* | 0.45\*\* | | **0.66\*\*** | **0.70\*\*** | | 0.53\*\* | 0.50\*\* | |
| $PM_{2.5}$ mass | 0.56\*\* | 0.24\*\* | 0.55\*\* | 0.23\* | 0.19 | 0.49\*\* | **0.70\*\*** | 0.36\*\* | 0.54\*\* |
| OC | 0.50\*\* | 0.10 | 0.55\*\* | 0.16 | 0.03 | 0.40\*\* | 0.61\*\* | 0.26\*\* | 0.50\*\* |
| EC | 0.49\*\* | 0.13\* | 0.51\*\* | 0.11 | 0.01 | 0.39\*\* | 0.60\*\* | 0.28\*\* | 0.44\*\* |
| WSOC | 0.55\*\* | 0.19\*\* | 0.52\*\* | 0.24\* | 0.20 | 0.41\*\* | **0.66\*\*** | 0.38\*\* | 0.54\*\* |
| BrC | 0.36\*\* | 0.04 | 0.41\*\* | 0.33\*\* | 0.14 | 0.51\*\* | 0.51\*\* | 0.33\*\* | **0.69\*\*** |
| $SO_4^{2-}$ | 0.41\*\* | 0.37\*\* | 0.34\*\* | 0.42\*\* | 0.35\*\* | 0.48\*\* | 0.33\*\* | 0.27\*\* | 0.18\* |
| WS-K | 0.49\*\* | 0.30\*\* | 0.50\*\* | 0.32\*\* | 0.08 | 0.46\*\* | 0.52\*\* | 0.33\*\* | 0.56\*\* |
| WS-Fe | 0.47\*\* | 0.17\*\* | 0.50\*\* | 0.43\*\* | 0.21 | 0.59\*\* | 0.52\*\* | 0.24\*\* | 0.48\*\* |
| WS-Cu | 0.50\*\* | **0.74\*\*** | 0.36\*\* | 0.51\*\* | **0.79\*\*** | **0.65\*\*** | 0.60\*\* | **0.78\*\*** | 0.31\*\* |
| WS-Mn | 0.38\*\* | 0.19\*\* | 0.37\*\* | 0.34\*\* | 0.13 | 0.61\*\* | 0.49\*\* | 0.28\*\* | 0.37\*\* |
| WS-Zn | 0.31\*\* | 0.31\*\* | 0.34\*\* | 0.32\*\* | 0.11 | 0.46\*\* | 0.41\*\* | 0.14 | 0.31\*\* |

Note: \*\*p-value<0.01; \*p-value<0.05. Correlations not statistically significant (p>0.05) are in grey, r>0.65 are bold. All metals listed are water-soluble metals.

**Table 2. Multivariate linear regression models for OP metrics.**

| | Fe | Cu | WSOC | BrC | metal–organic | | metal–metal | | intercept | $R^2$ |
|---|---|---|---|---|---|---|---|---|---|---|
| $OP^{DTT}$ | 2.28E-3 | 2.69E-3 | | 5.75E-2 | -1.36E-3 | Cu\*BrC | -4.09E-05 | Fe\*Cu | 0.13 | 0.40 |
| $OP^{AA}$ | 1.17E-1 | 1.07E-1 | 9.32E-1 | | -1.30E-2 | Fe\*WSOC | -2.05E-3 | Fe\*Cu | 4.38 | 0.54 |
| $OP^{GSH}$ | | 7.41E-2 | | 2.77E-1 | | | | | 3.65 | 0.56 |

Note: All metals are water-soluble metals. The values represent the coefficients for variables. Cell is left blank where the corresponding variable is not included in the equation. As an example, the linear equation of $OP^{DTT}$ is as follows: $OP^{DTT} = 2.28E\text{-}3*Fe + 2.69E\text{-}3*Cu + 5.75E\text{-}2*BrC - 1.36E\text{-}3*(Cu*BrC) - 4.09E\text{-}5*(Fe*Cu) + 0.13$. The concentrations of water-soluble PM components are in units of μg m$^{-3}$, whereas the unit for BrC is Mm$^{-1}$.






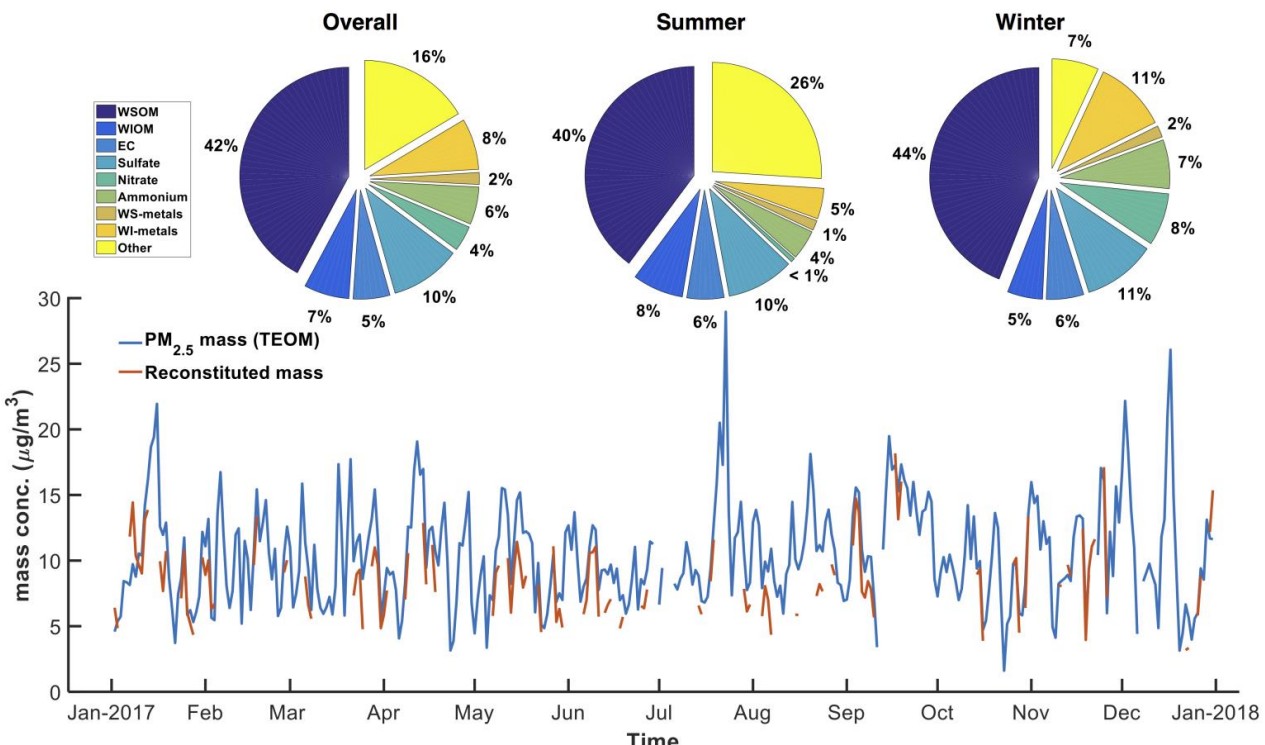

Figure 1: Time series of PM$_{2.5}$ mass concentration. The pie charts show the average aerosol composition based on PM$_{2.5}$ mass measured by the TEOM during the whole sampling year, summer and winter. WSOM: water-soluble organic matter (=WSOC*1.6); WIOM: water-insoluble organic matter (=OM-WSOM); WS-metals: sum of water-soluble metals, including Al, Mg, Ca, K, Fe, Cu, Mn, Zn; WI-metals: water-insoluble metals (=total_metals-WS_metals). Summer: Jun–Aug; winter: Jan–Feb and Nov–Dec.


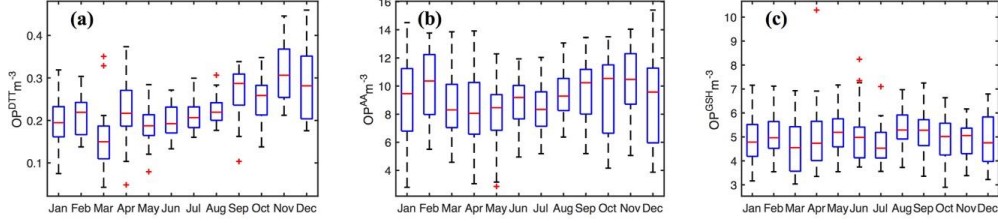

Figure 2: Temporal variation for (a) OP$^{DTT}$ m$^{-3}$ (nmol min$^{-1}$ m$^{-3}$), (b) OP$^{AA}$ m$^{-3}$ (% depletion of AA m$^{-3}$), and (c) OP$^{GSH}$ m$^{-3}$ (% depletion of GSH m$^{-3}$).





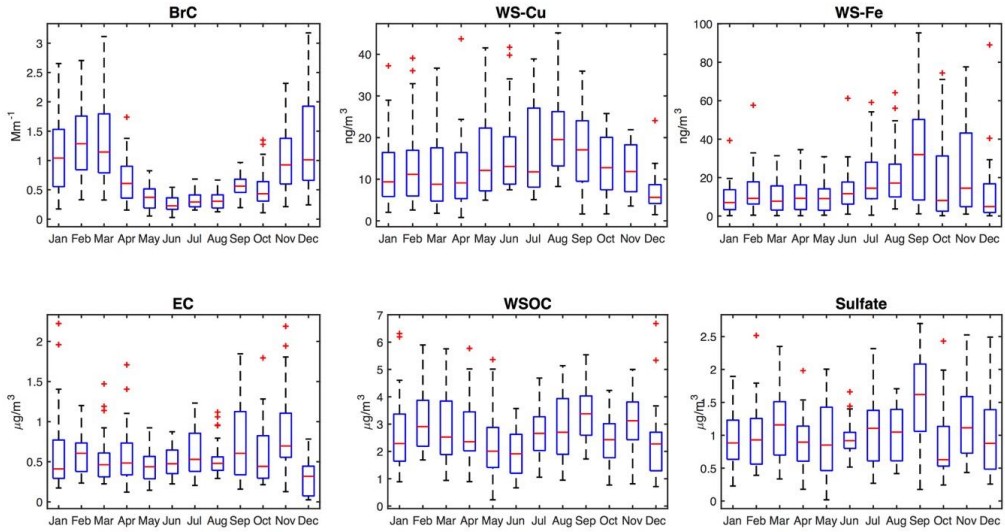


**Figure 3: Temporal variation for select PM species.**

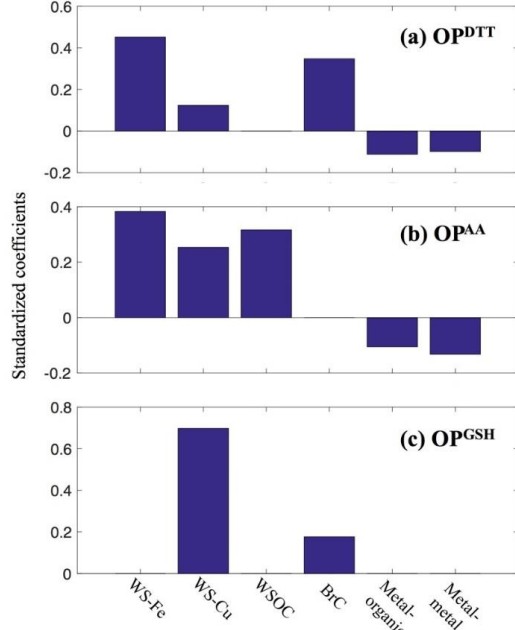

**Figure 4: Standardized regression coefficients for different OP measures with selected PM components.**