# Peer review of "Characterization and comparison of $PM_{2.5}$ oxidative potential assessed by two acellular assays"

_Atmospheric Chemistry and Physics, 2019_

## Referee Comment (RC1) · Anonymous Referee #1 · 28 Nov 2019

This manuscript showed the chemical composition and oxidative potentials (OP) of fine particulate matter (PM$_{2.5}$) in Atlanta at a year-long time scale. Moreover, the authors investigated the correlation of probe-based aerosol OP with abundance of different PM constituents. They found that dithiothreitol- and ascorbic acid-based OP exhibited moderate correlation with the abundance of water-soluble transition metals (Fe and Cu) and organic compounds (WSOC and brown carbon), whereas the glutathione (GSH)-based OP showed strong correlation with the water-soluble Cu. Finally, the authors developed a multivariate linear regression model to evaluate the plausible contributions of metals, organic compounds, metal–organic and metal–metal interactions to aerosol OP. Overall the topic is interesting. The manuscript cannot be published in its current form, but it may be publishable in *Atmos. Chem. Phys.* if the following comments can be thoroughly responded in the revised paper,

1. What is the atmospheric implications of aerosol OP, which may merit the current work to be publishable in *ACP* rather than an aerosol or air pollution health related journal? Some relevant discussions may be needed in the section 1 or 4.

2. The authors mainly described the correlations of different aerosol constituents with the OP reflected by different types of acellular assays. However, the manuscript lacks discussions and insight into the underlying chemical mechanisms of the interactions among different probes and PM constituents in water or the synthetic respiratory tract lining fluid.

3. In the manuscript especially the Figure 2, the authors only showed the OP values in the unit of nmol/min/m$^3$, which strongly associates with PM$_{2.5}$ concentrations. In contrast, the OP values in the unit of nmol/min/µg may exhibit stronger correlation with PM$_{2.5}$ composition. Therefore, the authors should present and discuss the OP values in the unit of nmol/min/µg as well as their dependence on different types of OP assays.

4. The Figures 1 and 3 are related to the chemical composition of PM$_{2.5}$, and the Figures 2 and 4 are for aerosol OP. Thus, it may be more suitable to present the current Figure 3 as Figure 2, and the current Figure 2 as Figure 3.

5. What is the association of water insoluble organic matter and metals in PM$_{2.5}$ (in Figure 1) with probe-based aerosol OP?

6. L158-172: whether the efficiency of NAPDH and GR to reduce GS-TNB to GSH can be interfered by the co-existence of ascorbic acid? Similarly, to which extent the co-variation of ascorbic acid and GSH concentrations will influence the OP$^{AA}$ and OP$^{GSH}$?

7. L243-244: The sentence of 'However, they could be considered as indicators of other compounds simultaneously produced by the same source' is a vague statement, which needs further clarification.

8. L249: What does the 'PM species' exactly refer to?

9. L274 (3.3 Temporal variation): to discuss the seasonal distribution of OP clearly, the averaged $PM_{2.5}$ OP of different seasons should be presented in Figures 2, 3 or SI, similar like the seasonal distribution of different PM components in Figure 1.

10. L281: Except for aerosol composition, the concentration of $PM_{2.5}$ and size distribution of redox active $PM_{2.5}$ constituents may also influence the seasonal distribution of OP (Lyu et al., Environ. Sci. Technol. 2018, 52, 6592-6600). Thus, the temporal variation of $PM_{2.5}$ OP should be discussed deeply.

11. L305-313: It has been found that secondary organic aerosols-bound water-soluble substances such as organic peroxide, highly oxygenated molecules, and semiquinone radicals etc. are redox active in producing reactive oxygen species through reactions with water, antioxidants, or lung cells (Khachatryan et al. Environ. Sci. Technol. 2011, 45, 19, 8559-8566; Tong et al., Environ. Sci. Technol. 2018, 52, 11642-11651; Tong et al., Environ. Sci. Technol. 2019, 53, 12506-12518; Zhou et al., Atmos. Chem. Phys. Discuss., https://doi.org/10.5194/acp-2019-190; Chowdhury et al., Environ. Sci. Technol. Lett. 2018, 5, 424−430; Chowdhury et al., Environ. Sci. Technol. 2019, DOI: 10.1021/acs.est.9b04449), thus the contribution or connection of these and other relevant WSOC substances to the OP of $PM_{2.5}$ should be discussed properly.

12. The y-axis title of the upper left panel (for BrC) in Fig. 3 should be corrected.

---

## Referee Comment (RC2) · Anonymous Referee #2 · 17 Dec 2019

In this work, the authors compared the results from 3 different acellular assays of oxidative potential in 2 different media. OP has recently become a popular topic of research due to its potential to represent PM's ability to drive oxidative stress and explain PM health effects. Understanding the assays used to measure OP is an important topic for atmospheric chemists, because they will provide insights into sources and/or compounds that may be particularly toxic. The authors found different level of sensitivities of these assays to different components, such as copper, iron, and organic compounds. These relationships were investigated by association, using multilinear regression models. Overall the results are a valuable contribution and are complementary to those currently in the literature. I just one major point of concern, and I hope the authors will consider it while revising the manuscript. I recommend publication in ACP

after considering these questions/comments.

My major issue with this work is the use of a per-air volume measure of OP (extrinsic OP) rather than a per-PM mass measure. All the comparisons made here are chemical, with the attempt to associate a particular fraction of PM to its contribution to OP. In that case, I would argue that the OP should be an intrinsic measure (i.e. oxidant depletion rate per PM mass). Otherwise the variability could be driven by total PM mass. I understand that the assays were performed on a per filter basis (which is equivalent to a per-volume basis), and it might be difficult to fix the amount of PM mass used to analyze OP. At the very least, there needs to be a discussion examining whether or not the variability in OP (and therefore the reported associations shown here) is driven by the PM mass, rather than its composition.

Other minor comments: 1. Does RTLF composition change with different regions in the lung? Given the sensitivity of the assay results to the relative concentrations of AA and GSH, this may be important. (This may seem like an obvious question to medical researchers or toxicologist, but an atmospheric audience for ACP might not understand.)

2. Samples are collected on a daily basis. Would that bias against sources that vary on shorter timescales (i.e. traffic-related emissions of metals)? If so, that should be stated as a limitation of this study.

3. Should we really expect a difference between summer and winter, given that the climate in Atlanta is similar between the seasons? What are the known differences in the sources between summer and winter this area? This type of comparison can be somewhat misleading and is likely not generalizable to other regions, because every city might have its own characteristic summer/winter sources. Just seeing "summer" and "winter" in analysis, one could jump to the wrong conclusions.

3. It would be useful to state in the Methods section the concentration of PM during these assays. PM concentrations should be much lower than those of the antioxidants

to ensure one is looking at the catalytic redox cycling.

4. Limits of detection and quantification for all of the assays should be reported.

5. The discussion around BrC comparison needs to be better motivated. It is not clear why that comparison was made in the first place, other than that measurement was available and it was convenient to make that comparison. BrC from biomass burning, for example, can be derived from nitrophenols, and is not exclusively HULIS. Unlike the other chemical species, BrC is not chemically defined, but rather an optically defined group of compounds, so their contribution to OP might not be straightforward.

6. Why is EC not included in the multilinear regression analysis? It seems to have a reasonable Pearson's r from Table 1.

Technical/formatting comments:

Line 164: typo after GR Line 169: typo in 2-vinylpyridine; not sure if the abbreviation 2-VP is needed if it is not used again Line 185: replace "required" with "performed" Line 257: If UA is not studied here, it might be better not to include UA in this comparison Line 266: "consistent lower" should be "consistently lower" Line 365: "shown the strongest estimated effect" is a strange word choice. Perhaps "estimated to have the strongest effect"? Line 388-390: The sentence here is stylistically awkward and grammatically incorrect. Table 2: the number of digits in the exponent are not consistent (some are E-3, and some are E-05)
* * *

---

## Author Comment (AC1) · 10 Feb 2020

We thank the reviewers for their time and comments. Below are detailed responses to each comment. The responses are italicized, and the modified texts are in red.

Response to anonymous referee #1 comments:

**This manuscript showed the chemical composition and oxidative potentials (OP) of fine particulate matter (PM2.5) in Atlanta at a year-long time scale. Moreover, the authors investigated the correlation of probe-based aerosol OP with abundance of different PM constituents. They found that dithiothreitol- and ascorbic acid-based OP exhibited moderate correlation with the abundance of water-soluble transition metals (Fe and Cu) and organic compounds (WSOC and brown carbon), whereas the glutathione (GSH)-based OP showed strong correlation with the water-soluble Cu. Finally, the authors developed a multivariate linear regression model to evaluate the plausible contributions of metals, organic compounds, metal–organic and metal–metal interactions to aerosol OP. Overall the topic is interesting. The manuscript cannot be published in its current form, but it may be publishable in *Atmos. Chem. Phys.* if the following comments can be thoroughly responded in the revised paper,**

1. **What is the atmospheric implications of aerosol OP, which may merit the current work to be publishable in *ACP* rather than an aerosol or air pollution health related journal? Some relevant discussions may be needed in the section 1 or 4.**

   *This paper deals with contrasting relatively new measures of atmospheric aerosol chemical characteristics and is of interest to the atmospheric chemistry community. In the last few years there have been numerous papers published in ACP on aerosol oxidative potential. Our work is motivated by the fact that aerosol OP has been linked to adverse health effects, but our focus here is on how two measures of OP in current use differ in their relationship to aerosol chemical composition.*

2. **The authors mainly described the correlations of different aerosol constituents with the OP reflected by different types of acellular assays. However, the manuscript lacks discussions and insight into the underlying chemical mechanisms of the interactions among different probes and PM constituents in water or the synthetic respiratory tract lining fluid.**

   *Elucidating the underlying chemical mechanisms is beyond the scope of this paper and specific chemical exposure–response relations were not explored in this study. However, possible mechanisms in the published literature were discussed in the paper as an explanation of our OP results. For example, we explained the antioxidants depletion in RTLF based on reactivity hierarchy of antioxidants and ligand speciation.*

3. **In the manuscript especially the Figure 2, the authors only showed the OP values in the unit of nmol/min/m3, which strongly associates with PM2.5 concentrations. In contrast, the OP values in the unit of nmol/min/μg may exhibit stronger correlation with PM2.5 composition. Therefore, the authors should present and discuss the OP values in the unit of nmol/min/μg as well as their dependence on different types of OP assays.**

*We agree that the mass-normalized OP could eliminate the collinearity in the correlations due to PM mass and does reflect the intrinsic property of PM resulting from PM composition. However, volume-normalized OP represents the human exposure to redox-active species, and thus is more health-related and useful in interpreting OP– health associations. Furthermore, compared to mass-normalized OP, the levels of volume-normalized OP provide more dynamic range useful for identifying OP-related species or sources. For example, the correlation of volume-normalized OP with BrC and K may suggest the influence of biomass burning, even though the actual drivers of OP metrics are confounded.*

*Correlation results between mass-normalized OP and species mass fraction have been added in SI (Table. S4). The results do not exhibit stronger correlation between intrinsic OP measures and PM composition, which may be due to nonlinear behavior of species with antioxidant oxidation and the interactions among species.*

4. **The Figures 1 and 3 are related to the chemical composition of PM2.5, and the Figures 2 and 4 are for aerosol OP. Thus, it may be more suitable to present the current Figure 3 as Figure 2, and the current Figure 2 as Figure 3.**

*Thank you for this suggestion. The figures were numbered according to their sequence in the text. The primary aim of this work was to compare the OP measures so we presented the temporal variation of OP first, and the PM species provided an explanation for the observed differences in OP measures.*

5. **What is the association of water insoluble organic matter and metals in PM2.5 (in Figure 1) with probe-based aerosol OP?**

*In another paper from this study (Gao et al., 2020, submitted to Atmos Environ), in which we focused more on the OP (DTT assay only) contributions from water-insoluble PM components, the association of total or water-insoluble PM species with total PM OP was studied in detail. However, the manuscript is focused only on the water-soluble OP fraction. Compared to the association between water-soluble species and OP measures, the association of water-insoluble PM species with water-soluble OP was less informative about determinants of water-soluble OP, and thus was not discussed in this paper.*

*To clarify, the manuscript has been modified.*

Line 276-277: "The OP (DTT assay only) contribution from water-insoluble components were discussed in detail in another paper from this study (Gao et al., accepted)."

6. **L158-172: whether the efficiency of NAPDH and GR to reduce GS-TNB to GSH can be interfered by the co-existence of ascorbic acid? Similarly, to which extent the co-variation of ascorbic acid and GSH concentrations will influence the OPAA and OPGSH?**

*The reviewer is thanked for raising this concern. The presence of AA is not expected to interfere with the reduction of GSSG or GS-TNB to GSH. This reduction reaction is the*

*key reaction involved in both total and oxidized glutathione measurements. We can check the reduction efficiency by examining the amount of total glutathione (GSx). Within error, the GSx concentration we measured in each plate well is consistent with the expected initial concentration (~200 μM per well), and the amount remains constant during the whole incubation, suggesting that all oxidized form glutathione can be reduced efficiently in GSx determination.*

*This study only indicated the antioxidants depletion in RTLF was affected by the RTLF composition. To what extent $OP^{AA}$ and $OP^{GSH}$ are influenced by RTLF composition still needs further investigation.*

7. **L243-244: The sentence of 'However, they could be considered as indicators of other compounds simultaneously produced by the same source' is a vague statement, which needs further clarification.**

*To improve clarity, the manuscript has been modified.*

Line 290-293: "However, the correlations may indicate the emission sources (e.g., as source tracers – vehicular emissions for EC, secondary processing for WSOC and $SO_4^{2-}$, and biomass burning for BrC and K), which also probably emit the water-soluble species contributing to the measured OP."

8. **L249: What does the "PM species" exactly refer to?**

*To clarify, we modified the manuscript.*

Line 298: "The associations found in this study between $OP^{DTT}$ and PM composition are consistent with a number of previous studies (Fang et al., 2016; Fang et al., 2015b; Verma et al., 2014; Yang et al., 2014), though the correlations in our work were weaker…"

9. **L274 (3.3 Temporal variation): to discuss the seasonal distribution of OP clearly, the averaged PM2.5 OP of different seasons should be presented in Figures 2, 3 or SI, similar like the seasonal distribution of different PM components in Figure 1.**

*Thank you for this suggestion. Figure 2 has been modified to include the averaged OP levels during warm and cold periods. The seasonal averaged PM species concentrations have been added into SI (Fig. S1). We have also modified the manuscript accordingly.*

Fig. 2 caption: "Temporal variation for (a) $OP^{DTT}$ $m^{-3}$ (nmol $min^{-1}$ $m^{-3}$), (b) $OP^{AA}$ $m^{-3}$ (% depletion of AA $m^{-3}$), and (c) $OP^{GSH}$ $m^{-3}$ (% depletion of GSH $m^{-3}$). Warm period: May–Aug; cold period: Jan–Feb and Nov–Dec."

Line 330: "The time series of the monthly and seasonal averages of different OP measures are shown…"

Line 337-338: "From the temporal variation of the OP-associated species shown in Fig. 3 (the seasonal averaged concentrations were given in Fig. S1), BrC had an obvious

seasonality… Water-soluble Cu is slightly higher in mid-summer (Aug) and water-soluble Fe is slightly higher in fall (Sep), but…"

10. **L281: Except for aerosol composition, the concentration of PM2.5 and size distribution of redox active PM2.5 constituents may also influence the seasonal distribution of OP (Lyu et al., Environ. Sci. Technol. 2018, 52, 6592-6600). Thus, the temporal variation of PM2.5 OP should be discussed deeply.**

*We agree with the reviewer that oxidative potential will vary across PM size fractions which may exhibit seasonal variance. Evaluation of multiple size fractions for PM and chemical composition was beyond the scope of this study. This study was solely based on PM$_{2.5}$ measurements with no further size distribution information, however, the change of PM composition resulted from seasonal size distribution shifts could be captured in our sampling. Either PM mass concentration or size distribution of PM constituents influence PM$_{2.5}$ OP by changing the composition of PM$_{2.5}$. Therefore, the PM OP essentially is still related to aerosol composition. This is also consistent with the findings in the reference cited by the reviewer, which demonstrated that the averaged OP levels for all size fractions were significantly correlated with PM redox active species (quinone and water-soluble metals).*

11. **L305-313: It has been found that secondary organic aerosols-bound water-soluble substances such as organic peroxide, highly oxygenated molecules, and semiquinone radicals etc. are redox active in producing reactive oxygen species through reactions with water, antioxidants, or lung cells (Khachatryan et al. Environ. Sci. Technol. 2011, 45, 19, 8559-8566; Tong et al., Environ. Sci. Technol. 2018, 52, 11642-11651; Tong et al., Environ. Sci. Technol. 2019, 53, 12506-12518; Zhou et al., Atmos. Chem. Phys. Discuss., https://doi.org/10.5194/acp-2019-190; Chowdhury et al., Environ. Sci. Technol. Lett. 2018, 5, 424−430; Chowdhury et al., Environ. Sci. Technol. 2019, DOI: 10.1021/acs.est.9b04449), thus the contribution or connection of these and other relevant WSOC substances to the OP of PM2.5 should be discussed properly.**

*Based on the reference cited by the reviewer, the redox active substances the reviewer mentioned are more likely bounded on particles in atmospheric conditions rather than WSOC. Given that we were only measuring water-soluble OP of long-time integrated PM samples, we may fail to capture the effects of these short-lived species or radicals. Therefore, including these substances as part of WSOC-related OP contribution may not be appropriate.*

*We have included some of these substances in the introduction to provide a more comprehensive description about OP contributors.*

Line 58-64: "The ability of PM to generate ROS *in vivo,* referred to as the oxidative potential (OP) of particles, has gained increasing attention as possibly a more integrative health-relevant measure of ambient PM toxicity than PM mass concentration which may contain a mix of highly toxic (e.g. polycyclic aromatic hydrocarbons (PAHs), quinones, environmentally persistent free radicals, highly oxygenated organic molecules, and transition metals) to relatively benign (e.g. sulfate and ammonium nitrate) PM

components (Frampton et al., 1999; Khachatryan et al., 2011; Lippmann, 2014; Tong et al., 2018; Tong et al., 2019).”

**12. The y-axis title of the upper left panel (for BrC) in Fig. 3 should be corrected.**

*Thank you for pointing this out! The y-axis title for BrC in Fig. 3 has been corrected from “$Mm^{-1}$” to “1/Mm” so that it remains consistent with the format of other species' units.*

**Reference:**

Gao, D., Mulholland, J. A., Russell, A. G., and Weber, R. J.: Characterization of water-insoluble oxidative potential of PM2.5 using the dithiothreitol assay, Atmos Environ (accepted)

Response to anonymous referee #2 comments:

**In this work, the authors compared the results from 3 different acellular assays of oxidative potential in 2 different media. OP has recently become a popular topic of research due to its potential to represent PM's ability to drive oxidative stress and explain PM health effects. Understanding the assays used to measure OP is an important topic for atmospheric chemists, because they will provide insights into sources and/or compounds that may be particularly toxic. The authors found different level of sensitivities of these assays to different components, such as copper, iron, and organic compounds. These relationships were investigated by association, using multilinear regression models. Overall the results are a valuable contribution and are complementary to those currently in the literature. I just one major point of concern, and I hope the authors will consider it while revising the manuscript. I recommend publication in ACP**

**My major issue with this work is the use of a per-air volume measure of OP (extrinsic OP) rather than a per-PM mass measure. All the comparisons made here are chemical, with the attempt to associate a particular fraction of PM to its contribution to OP. In that case, I would argue that the OP should be an intrinsic measure (i.e. oxidant depletion rate per PM mass). Otherwise the variability could be driven by total PM mass. I understand that the assays were performed on a per filter basis (which is equivalent to a per-volume basis), and it might be difficult to fix the amount of PM mass used to analyze OP. At the very least, there needs to be a discussion examining whether or not the variability in OP (and therefore the reported associations shown here) is driven by the PM mass, rather than its composition.**

*The reviewer is thanked for the comment, correlation results between intrinsic OP and species mass fraction have been added in SI. As we addressed in the response to reviewer #1, volume-normalized OP can reflect the human exposure to redox-active species and help explain the observed OP–health associations. OP per mass, however, is the intrinsic property of PM which is more applicable to studies of emissions. Since the goal of this study is to compare OP assays that have been found in health studies to be associated with adverse health effects, and to investigate the underlying chemical species that may be driving these associations, we focus on the OP per volume air.*

**Other minor comments:**

**1. Does RTLF composition change with different regions in the lung? Given the sensitivity of the assay results to the relative concentrations of AA and GSH, this may be important. (This may seem like an obvious question to medical researchers or toxicologist, but an atmospheric audience for ACP might not understand.)**
*Yes, RTLF composition changes with different regions in the lung. Based on the reviewer's comment, this point has been explicitly noted in the revised manuscript.*

Line 469-473: "There are marked differences in RTLF composition in different levels of respiratory tract. The synthetic RTLF reflects select antioxidants in the lung and other key constituents are not represented in this simplified chemical model. The DTT assay is also subject to similar limitations that DTT cannot fully represent the biological complexity. However, these assays can be used as PM screening tools and provide rapid health-relevant assessment of PM."

**2. Samples are collected on a daily basis. Would that bias against sources that vary on shorter timescales (i.e. traffic-related emissions of metals)? If so, that should be stated as a limitation of this study.**

*Thanks for this suggestion. This limitation has been added in the manuscript.*

Line 465-467: "We should note that the filter samples analyzed in this study were averaged over 24 hours, which may dampen variability in the emission sources that contribute to OP and obscure the impact of specific species (e.g., traffic-related metals) on redox activity of PM."

**3. Should we really expect a difference between summer and winter, given that the climate in Atlanta is similar between the seasons? What are the known differences in the sources between summer and winter this area? This type of comparison can be somewhat misleading and is likely not generalizable to other regions, because every city might have its own characteristic summer/winter sources. Just seeing "summer" and "winter" in analysis, one could jump to the wrong conclusions.**

*We thank the reviewer for the comment. While temperature change across seasons in Atlanta (typically varies from 35 °F to 89 °F) is not as extreme as other cities, there are notable differences. Consequentially, the seasonal difference in OP are expected. In previous studies conducted in the same region, obvious seasonal patterns were observed for water-soluble $OP^{DTT}$ and its related sources (Bates et al., 2015; Verma et al., 2014). The results suggested that $OP^{DTT}$ levels in Atlanta were generally higher in the cold months, driven mostly by biomass burning emissions, than in summer when secondary oxidation processes dominated $OP^{DTT}$. The temporal variation in our BrC data (Fig. 3) also supported the seasonally varying emission sources. Moreover, based on our correlation analysis, the association of OP measures with PM species differed by seasons, suggesting possible seasonal differences in OP metrics. Even though the temporal variation in OP from this study was not evident, we believe it is worthwhile to do such seasonality analysis as it may provide a better understanding of the impact of seasonally varying sources or species upon OP metrics.*

*To avoid misleading our readers, we have added a reminder in the revised manuscript.*

Line 467-469: "Furthermore, all these results were obtained from a specific location in Atlanta and should be interpreted and generalized with caution as the chemical composition or sources of PM varies by region."

**4. It would be useful to state in the Methods section the concentration of PM during these assays. PM concentrations should be much lower than those of the antioxidants to ensure one is looking at the catalytic redox cycling.**

*Thanks for the suggestion. The PM concentration in the water extracts has been specified in the manuscript.*

Line 168: "In brief, the PM extract (3.5 mL; $40\pm15$ µg mL$^{-1}$ of PM) was incubated with DTT solution…"

Line 189: "PM water extracts ($35\pm13$ µg of PM per mL) were transferred into a 96-well microplate with 180 µL of sample liquid in each well."

**5. Limits of detection and quantification for all of the assays should be reported.**

*The manuscript has been modified accordingly.*

In section 2.2.1 DTT assay, Line 183-184: "The limit of detection (LOD), defined as three times of the standard deviation of OP$^{DTT}$ for blanks, is 0.31 nmol min$^{-1}$...."

In section 2.2.2 RTLF assay, Line 216-217: "The LOD for AA and GSH depletion after 4 h incubation was 4.0 % and 4.5 %, respectively..."

**6. The discussion around BrC comparison needs to be better motivated. It is not clear why that comparison was made in the first place, other than that measurement was available and it was convenient to make that comparison. BrC from biomass burning, for example, can be derived from nitrophenols, and is not exclusively HULIS. Unlike the other chemical species, BrC is not chemically defined, but rather an optically defined group of compounds, so their contribution to OP might not be straightforward.**

*To better motivate the BrC vs. WSOC comparison, we have modified the manuscript.*

Line 362-365: "All models captured the contributions from organic species, however, the organic contributions in different models were represented by different measures of organics. In the OP$^{DTT}$ and OP$^{GSH}$ models, the organic contribution was denoted by BrC, whereas WSOC was used in the OP$^{AA}$ model. Although WSOC and BrC were correlated with each other (Table S1–S3), there is a difference between these two parameters."

*Regarding the reviewer's 2$^{nd}$ point, the reviewer is correct that BrC likely covers a wide range of aerosol components, which include but are not limited to HULIS. HULIS has been recognized as important components of BrC and has been found to be strongly linked to BrC (Hoffer et al., 2006; Laskin et al., 2015). As mentioned in the manuscript, Verma et al. (2012) passed the water extracts of PM$_{2.5}$ collected at Jefferson Street (the same location as in this study) through C-18 solid extraction columns which were commonly used for HULIS isolation, and they found that roughly 70–90 % of BrC was retained in hydrophobic fraction (i.e., HULIS). Therefore, it is*

*reasonable to denote the influence of HULIS on OP with BrC in this study. Furthermore, although there is evidence that BrC components may directly contribute to OP (Chen et al., 2019), the presence of BrC in the regression analysis may not imply their direct contribution to OP, but to indicate the possible influence of BrC-denoted aerosol components. It is true that BrC, as an optically defined measure, differs from other PM species in units and scales, which makes it difficult to inter-compare the OP contribution induced by each term. However, this problem was resolved by applying standardized regression, as noted in this paper.*

*To clarify, we have added more explanation of the BrC-related OP in the revised manuscript.*

Line 366-374: "…there is a difference between these two parameters. The OP contribution from BrC, an optically defined measure, may not be straightforward, however, the optical properties of BrC are related to chemical properties. There is evidence showing that BrC components may directly contribute to OP (Chen et al., 2019). It has been found that BrC predominantly represents the hydrophobic organic fraction…"

**7. Why is EC not included in the multilinear regression analysis? It seems to have a reasonable Pearson's r from Table 1.**

*This manuscript was focused on water-soluble OP which we assume is more driven by water-soluble PM components. Therefore, in the multivariate regression analysis, we used water-soluble species as input to construct the models. Furthermore, EC was also correlated with some of the selected water-soluble species, such as WSOC and water-soluble Mn and Fe (Table S1). In this instance, even if EC was chosen as one of the predictors, the stepwise regression procedure would identify it as a redundant predictor and exclude it from the model. To clarify, the manuscript has been modified.*

Line 344-349: "Given that one or more PM components contributed to these measures of OP, multivariate linear regression analysis was conducted to identify the main  PM components that drive the variability in OP and provide a contrast between the assays. Water-soluble organic species (WSOC or BrC) and metals, mainly Fe, Cu, Mn, were selected as the independent variables to form multivariate linear regression models for $OP^{DTT}$ and $OP^{AA}$, based on their high correlations, as noted above. EC, though also correlated with $OP^{DTT}$ and $OP^{AA}$, was not chosen as one of the predictors due to its correlations with selected water-soluble species (e.g., WSOC, water-soluble Mn and Fe)."

**Technical/formatting comments:**

**Line 164: typo after GR**

*We made a double-check and found no typo here.*

**Line 169: typo in 2-vinylpyridine; not sure if the abbreviation 2-VP is needed if it is not used again**

*Typo has been corrected, and the abbreviation was deleted.*

Line 208: "5 µL of 2-vinylpyridine was added…"

**Line 185: replace "required" with "performed"**

*Corrected.*

Line 227: "For the analysis of water-soluble metals, no digestion was performed."

**Line 257: If UA is not studied here, it might be better not to include UA in this comparison**

*UA has been deleted in the comparison.*

Line 306: "...within the antioxidant model with GSH>AA (Zielinski et al., 1999)."

**Line 266: "consistent lower" should be "consistently lower"**

*Corrected.*

Line 315: "The consistently lower correlation…"

**Line 365: "shown the strongest estimated effect" is a strange word choice. Perhaps "estimated to have the strongest effect"?**

*The sentence has been modified.*

Line 435: "Water-soluble Fe, as the most important determinant of $OP^{DTT}$, has been estimated to have the strongest effect on cardiovascular outcomes in the Atlanta metropolitan region…"

**Line 388-390: The sentence here is stylistically awkward and grammatically incorrect.**

*The sentence has been modified.*

Line 462-463: "However, $OP^{DTT}$ and $OP^{AA}$ are more chemically integrative OP measures compared to $OP^{GSH}$, and thus may be more informative and helpful in linking OP with health end points..."

**Table 2: the number of digits in the exponent are not consistent (some are E-3, and some are E-05)**

*Corrected.*

[revised manuscript text omitted]

---

## Referee Report (RR1)

Review of "Characterization and comparison of PM$_{2.5}$ oxidative potential assessed by two acellular assays" by Dong Gao et al.

The manuscript was substantially improved. I only have minor comments.

1. L132, 193, 197, and others: you may want to keep the accuracy and unit of filter area values to be consistent.
2. Line 404: you mentioned 3 different assay metrics, but you frequently discussed as 2 in the title and main text. Please double check and make sure the definition and categorization are consistent.
3. It is better to explain what does the 'TEOM' mean in the caption of Figure 1.
4. Figure 3 caption: change the 'species' to 'components'?

---

## Author Response (AR2)

We really thank the editor for their comments. We have addressed the comments below, with the responses italicized and the modified texts in red.

1. L132, 193, 197, and others: you may want to keep the accuracy and unit of filter area values to be consistent.

*Thank you for the suggestion. To be consistent, the filter area values have been rounded to one decimal place and in units of $cm^2$.*

2. Line 404: you mentioned 3 different assay metrics, but you frequently discussed as 2 in the title and main text. Please double check and make sure the definition and categorization are consistent.

*To clarify, the sentence has been modified.*

Line 404, "This study suggests that all three OP metrics, $OP^{AA}$ and $OP^{GSH}$ in the RTLF assay as well as $OP^{DTT}$ in the DTT assay, are associated with transition metal ions."

3. It is better to explain what does the 'TEOM' mean in the caption of Figure 1.

*We have revised the caption of Fig. 1 according to the suggestion to include the explanation of TEOM.*

[revised manuscript text omitted]